# Technical Requirements for 2023 IMO GHG Strategy

**Chunchang Zhang** [1,2], **Jia Zhu** [1,2], **Huiru Guo** [1,2], **Shuye Xue** [2], **Xian Wang** [2,3], **Zhihuan Wang** [2,4], **Taishan Chen** [1,2], **Liu Yang** [1,2], **Xiangming Zeng** [1,2] and **Penghao Su** [2,5,*]

[1] Merchant Marine College, Shanghai Maritime University, Shanghai 201306, China; cczhang@shmtu.edu.cn (C.Z.); 202330110091@stu.shmtu.edu.cn (J.Z.); hrguo@shmtu.edu.cn (H.G.); tschen@shmtu.edu.cn (T.C.); liuyang@shmtu.edu.cn (L.Y.); xmzeng@shmtu.edu.cn (X.Z.)

[2] Ship Energy Efficiency Data Center, Shanghai Maritime University, Shanghai 201306, China; syxue@shmtu.edu.cn (S.X.); xianwang@shmtu.edu.cn (X.W.); zhwang@shmtu.edu.cn (Z.W.)

[3] College of Foreign Languages, Shanghai Maritime University, Shanghai 201306, China

[4] Institute of Logistics Science and Engineering, Shanghai Maritime University, Shanghai 201306, China

[5] College of Ocean Science and Engineering, Shanghai Maritime University, Shanghai 201306, China

* Correspondence: phsu@shmtu.edu.cn; Tel.: +86-21-38282535

**Abstract:** The 80th session of the IMO Maritime Environment Protection Committee (MEPC 80) adopted the 2023 IMO Strategy on the Reduction of GHG Emissions from Ships (2023 IMO GHG Strategy), with enhanced targets to tackle harmful emissions. This study strives to provide an exact interpretation of the target of the 2023 IMO GHG Strategy and reveal the technical requirements therein. Decarbonization targets were expressed in IMO GHG emission scenarios for specifications. Model calculations and parameterizations were in line with IMO GHG reduction principles and decarbonizing practices in the shipping sector to avoid the prejudicial tendency of alternative fuels and the overestimated integral efficiency of short-term measures in existing predictions. IMO DCS data were used for the first time to gain reliable practical efficiencies of newly adopted regulations and further reduce the model uncertainty. The results demonstrated that the decarbonization goals for emission intensity were actually 51.5–62.5% in the IMO GHG reduction scenarios, which was much higher than the IMO recommended value of 40% as the target. Combined with the continuous applications of short-term measures, onshore power and regulations were required to contribute their maximum potential no later than the year 2030. Even so, considerable penetration (15.0–26.0%) of alternative fuels will be required by 2030 to achieve the decarbonization goals in the 90% and 130% scenarios, respectively, both far beyond the expected value in the 2023 IMO GHG Strategy (i.e., 5–10%). Until 2050, decarbonization from alternative fuels is required to achieve ~95%. Sustainable biodiesel and LNG are the necessary choices in all time periods, while the roles of e-methanol and e-ammonia deserve to be considered in the long term. Our findings highlight the intense technical requirements behind the 2023 IMO GHG Strategy and provide a pathway option for a fair and impartial transition to zero GHG emissions in the shipping sector, which might be meaningful to policymakers.

**Keywords:** 2023 IMO GHG Strategy; decarbonization; technical requirement; alternative fuel

## 1. Introduction

In July 2023, the International Maritime Organization (IMO) adopted the '2023 IMO strategy on the reduction of GHG emissions from ships' (2023 IMO GHG Strategy) [1] during the 80th session of the IMO Maritime Environment Protection Committee (MEPC 80), which marked the decarbonization process in the shipping sector entering a medium–long-term stage. The 2023 IMO GHG Strategy indicates that GHG emissions from international shipping will reach net zero by or around 2050, which is more urgent than the target in the 2018 IMO GHG Strategy (50% by 2050) [2]. The importance of alternative fuel/energy, in particular, is emphasized in the 2023 IMO GHG Strategy. Moreover, the ambitious

levels of reduction and indicative checkpoints taking well-to-wake (WtW) emissions into account are addressed in the 'Guidelines on lifecycle GHG intensity of marine fuels' (LCA guidelines, also approved at MEPC 80) [3] to prevent the shifting of shipping emissions to other sectors.

The 2023 IMO GHG Strategy is expected to produce a further push toward decarbonization in the shipping sector. Actions are supposed to be in line with the strategy; thus, the requirements of the reduction ambitions must be revealed and understood appropriately from the very first step. Previous assessments and predictions [4] consistently employed the IMO's recommendation of at least a 40% reduction in emission intensity by 2030 [1,2] as a reduction target. Doing so could hardly cover all the requirements at indicative checkpoints in various scenarios. For example, 'emissions (from international shipping) could represent between 90% and 130% of 2008 emissions by 2050', 'to reduce the total annual GHG emissions from international shipping by at least 20%, striving for 30%, by 2030, compared to 2008', and 'to reduce the total annual GHG emissions from international shipping by at least 70%, striving for 80%, by 2040, compared to 2008' [1]. An appropriate understanding of the temporal and specific emission intensity targets in line with the shipping sector development predictions and reduction ambitions is crucial for assessments and predictions.

It is also challenging to design a comprehensive, goal-achieving pathway of the technical measures (that is, the technical requirements) for the whole shipping sector. Most of the GHG reduction potentials of the IMO's short-term techniques have been investigated individually, and tempting ideal results have been reported, e.g., vessel size optimization (47%) [5,6], hull shape optimization (15%) [7], lightweight materials (9%) [4], air lubrication (8%) [8], hull coating (5%) [9], propulsion efficiency devices (9.6%) [10], speed optimization (19%) [4,11], voyage optimization (12.5%) [12,13], ballast water reduction (7%) [4,14], etc. Nevertheless, there has recently been less optimism related to the onboard practices of these measures. In the Fourth IMO GHG Study 2020, the IMO stated that more than half of these improvements were achieved before 2012 and that the pace of carbon intensity reduction has been further slowing down since 2015, with average annual percentage changes ranging from 1 to 2% [15]. Unfortunately, individual potentials are frequently suggested in measure introductions/recommendations rather than integral practical potentials [16,17], which results in overestimated integral reduction efficiencies for these measures and an underestimation of the technical requirements of the reduction pathway prediction.

Moreover, the bias toward marine fuel applications should be avoided, e.g., the tendentious attention turning toward methanol, ammonia, and hydrogen fuels from readily available liquified natural gas (LNG) and biodiesel [18–23]. In principle, fuel diversity can reduce risk in the shipping sector, and one-sided emphasis on the advantages of a single fuel type is dangerous. In reality, methanol, ammonia, and hydrogen fuels do not exhibit convincing low emissions in the LCA stage compared to LNG and biodiesel [16,24]. In addition, it is anticipated that fuels produced by hydrogenating either biomass or carbon dioxide are likely the solution, regardless of the fuel type [25,26]. Consequently, all fuels are included in the framework of the 2023 IMO GHG Strategy and the LCA guidelines based on the principles of 'non-discrimination and no more favorable treatment, enshrined in MARPOL and other IMO conventions' and 'common but differentiated responsibilities and respective capabilities, in the light of different national circumstances.

Lastly, efforts should be made to reduce the complexity and uncertainties of the prediction model. The technical pathway to the GHG reduction target in the shipping sector comprises dozens of measures [15], and so does the prediction model [27]. Thanks to the IMO's comprehensive assessment of the integral carbon intensity reduction of short-term measures, the relevant calculation and parameterization can be simplified in the following prediction model. In a prediction based on the 2023 IMO GHG Strategy, attention can mainly be paid to innovative measures and regulations, such as the LCAs of zero or near-zero GHG emission technologies, fuels, and/or energy; onshore power;

the carbon intensity indicator (CII) rating; etc. Model uncertainties are mainly associated with the LCA emissions of fuel/energy feedstock and carbon capture. Because the LCA item is actually beyond the responsibilities of the IMO and the shipping sector, most of the parameters are null in the LCA guidelines and are under discussion. At present, using the LCA emissions of fuel/energy reported by relevant organizations is likely the best option to avoid uncertainties, as well as double counting, which is also in line with the strategy of the IMO Fuel GHG standard (GFS) (in development). In addition, ship energy efficiency data from the IMO Data Collection System (IMO DCS, since 2019) for fuel oil consumption favor a comprehensive assessment of the practical efficiencies of newly adopted regulations [9,28,29] and further reduce model uncertainties.

This study strove to provide an exact interpretation of the target of the 2023 IMO GHG Strategy and reveal the 'real' technical requirements in line with the IMO principles and sector practices. To this end, the target in the 2023 IMO GHG Strategy was specified in emission scenarios from the Fourth IMO GHG Study 2020. A concise, technical, pathway-predicting model was established by combining the efficiency improvement (EIIMO) of short-term measures predicted by the IMO with measures that will still work after the implementation of the 2023 IMO GHG Strategy, including onshore power (OP) usage, carbon intensity indicator (CII) rating promotion, and LCA zero-emission fuel application. Parameterizations referred to information from the fuel sectors and the 2020 IMO DCS data for sensible purposes. The results might be meaningful to policy and strategy makers in the shipping sector.

## 2. Data and Methods

### 2.1. IMO Ambitions and Indicative Checkpoints

The IMO ambitions and indicative checkpoints in the 2023 IMO GHG Strategy are summarized in Table 1, where emission intensity is defined as the emission per transport work. The targets in the emission amounts are specified in the form of emission intensity, and they were used as references in decarbonization pathway assessments.

**Table 1.** IMO decarbonization ambitions and indicative checkpoints.

| Year | Intensity | Amount | Zero- or Near-Zero-Emission Technologies and Fuels/Energy |
|------|-----------|--------|-----------------------------------------------------------|
| 2008 | baseline | baseline | n.r. |
| 2030 | by $\geq$40% | by 20–30% | by 5–10% |
| 2040 | n.r. | by 70–80% | n.r. |
| ~2050 | n.r. | net zero | net zero |

n.r., not regulated.

### 2.2. Scenarios

According to the Fourth IMO GHG Study 2020, emissions from shipping may represent between ~90% and ~130% of 2008 emissions by 2050, assuming that the shipping sector is in the business-as-usual (BAU) scenarios. The definition of BAU is that no new shipping regulations are adopted that impact emissions or energy efficiency. This shows how shipping emissions will develop if other sectors follow a certain economic and climate pathway and shipping does not. In the context of this study, analysis proceeded on the basis of the data in the 90% and 130% scenarios, as summarized in Tables 2 and 3, including efficiency improvements ($EI_{IMO}$) compared to 2018 and $CO_2$ emissions ($e_{CO2}$) and changes compared to 2008 (%$e_{2008}$).

**Table 2.** Projections in 90% scenario (SSP4_RCP2.6_G) [5].

| Year | *TW* (Billion Tonne·Miles) | *EI*$_{IMO}$ (%) | $e_{CO2}$ ($10^{10}$ Tonne) | %$e_{2008}$ |
|---|---|---|---|---|
| 2008 | 44,503 | Baseline | 1.110 | Baseline |
| 2018 | 59,230 | 0.0 | 0.999 | 90.0 |
| 2020 | 62,331 | 5.0 | 1.022 | 92.1 |
| 2025 | 68,305 | 9.0 | 1.047 | 94.3 |
| 2030 | 72,744 | 12.0 | 1.036 | 93.3 |
| 2035 | 76,570 | 14.0 | 1.032 | 93.0 |
| 2040 | 79,750 | 14.0 | 1.028 | 92.6 |
| 2045 | 82,162 | 14.0 | 1.037 | 93.4 |
| 2050 | 84,157 | 15.0 | 1.040 | 93.7 |

**Table 3.** Projections in 130% scenario (SSP2_RCP2.6_L) [5].

| Year | *TW* (Billion Tonne·Miles) | *EI*$_{IMO}$ (%) | $e_{CO2}$ ($10^{10}$ Tonne) | %$e_{2008}$ |
|---|---|---|---|---|
| 2008 | 44,503 | - | 1.110 | - |
| 2018 | 59,230 | 0.0 | 0.999 | 90.0 |
| 2020 | 62,518 | 1.0 | 1.026 | 92.4 |
| 2025 | 71,707 | 5.0 | 1.095 | 98.6 |
| 2030 | 82,460 | 10.0 | 1.154 | 104.0 |
| 2035 | 91,448 | 13.0 | 1.209 | 108.9 |
| 2040 | 101,604 | 15.0 | 1.295 | 116.7 |
| 2045 | 109,958 | 16.0 | 1.367 | 123.2 |
| 2050 | 119,429 | 17.0 | 1.456 | 131.2 |

The projection measures were introduced in the Fourth IMO GHG Study 2020. In brief, the projections were based on GDP and population projections from the Shared Socio-Economic Pathways (SSPs) developed by the Intergovernmental Panel on Climate Change (IPCC). SSP2 (in the 130% scenario) and SSP4 (in the 90% scenario) represented middle and unequal/divided global development paths, respectively. The SSPs were combined with representative concentration pathways (RCPs, also developed by the IPCC) for integral assessment. In the 90% and 130% scenarios, RCP2.6 represented very low GHG emissions in line with the goal of keeping the global mean temperature increase below 2 °C.

The logistic model (L, in the 130% scenario) and the gravity model (G, in the 90% scenario) were employed to project transport work. The difference between the two measures indicated the uncertainty inherent in making projections about future developments. The logistic model assumed that transport work was related to the world's total GDP. It accurately described the past for different cargo types and captured the specificities of the global transport of different commodities. Because it was based on global data, it did not capture the peculiarities of countries' bilateral trade flows.

By contrast, the gravity model analysis presumed that transport work was a function of the per capita GDPs and populations of the trading countries and used econometric techniques to estimate the elasticity of transport work with respect to its drivers based on panel data of bilateral trade flows. It used data on the volume of bilateral trade flows for a five-year period (2014–2018) and estimated the share of maritime transport in those trade flows to generate mode-specific trade volume data. It used panel data techniques to determine the elasticities of trade and accurately described how GDP and population variations impacted sea trade, capturing the idiosyncrasies of each trade flow.

The projected $e_{CO2}$ values in Tables 2 and 3 are integrations of *TW* and *EI*$_{IMO}$. *EI*$_{IMO}$ was projected as a result of changes in fleet composition (e.g., the replacement of smaller ships with larger ships, with higher demand growth for containers than for dry bulk ships and tankers), regulatory efficiency improvements (e.g., the replacement of pre-EEDI ships with EEDI Phase 1, 2, and 3 ships), and market-driven efficiency improvements. Scenarios with higher transport growth had a larger share of new ships in the fleet, which resulted in greater efficiency improvements.

As such, in this study, the total efficiency improvement was projected by combining the $EI_{IMO}$ predicted by the IMO in 2020 with the subsequent improvements derived from pathways such as the application of onshore power, operational CII rating regulations [30–33], and low-/zero-carbon fuels.

### 2.3. Ship Fuel Consumption Data

To investigate the current situation of global shipping energy efficiency and the decarbonization potential of actions, 2020 annual global ship fuel combustion data from the IMO DCS [34] were employed for analysis. All reports of international ships in 2020 with specified voyages were collected from the IMO DCS, and over 100,000 data were obtained. Crucial information included the ship type, deadweight tonnes, gross tonnes (GTs), sailing distance, fuel consumption, onshore power consumption, etc. Before analysis, the IMO DCS data were processed to rule out input errors by ship staff and, thus, the calculated values of the total fuel combustion in this study might be lower than those reported elsewhere. Accordingly, the IMO DCS data were only applied in regularity analysis, such as for the portions of onshore power consumption, fuel consumption by type, and operational CII ratings.

$CO_2$ emissions were determined using the fuel consumption and TtW transfer coefficients presented in the IMO documents [35]. Other greenhouse gases such as $CH_4$ and $N_2O$ were not counted due to a lack of emission information.

### 2.4. Model

In the scope of the entire shipping sector, the carbon intensity reduction potential was calculated by integrating the improvements of various measures. We made the following assumptions: (1) $EI_{IMO}$ predicted on the basis of the technical level until 2018 (see Section 2.2) could represent the integral decarbonization potential of short-term measures. (2) Subsequent measures including onshore power usage (OP), CII rating promotion (ProCII), and LCA zero-emission fuel and/or energy application (F) could make improvements simultaneously with the short-term measures and, thus, the decarbonization potential of the short-term and subsequent measures could be integrated. (3) A ship could choose OP, ProCII, LCA zero-emission fuel, or all of the measures to reduce its emissions. Thus, their decarbonization potential could be summed. In particular, ProCII could be achieved using all other measures, and probable double counting was solved using time-specific parameterizations (see principle d in Section 2.5).

Consequently, the total decarbonization potential of emission intensity ($Z_t$, %) could be expressed using Equation (1):

$$Z_t = 1 - (1 - EI_{IMO,t}) \times [(1 - Z_{OP,t} - Z_{ProCII,t} - Z_{F,t}) \times (1 - Z_{2020})] \tag{1}$$

where the subscript $t$ represents a calendar year. The square brackets emphasize that $Z_{OP,t}$, $Z_{ProCII,t}$, and $Z_{F,t}$ use $Z_{2020}$, rather than $Z_{2018}$, as a reference. This setting was reasonable considering that a series of regulations entered into force no earlier than 2023 and limited action on onshore power usage. CII promotion and fuel alteration were not performed before 2020. Equation (1) shows that these 3 measures do not exclude each other and can improve carbon intensity individually beyond $EI_{IMO}$. It should be specified that, in practice, ships promote their CII rating mainly via biodiesel usage due to the limit-promoting potential of measures included in $EI_{IMO}$; thus, $Z_{ProCII,t}$ and $Z_{biodiesel,t}$ could be parameterized by the time node, as interpreted in Section 2.5.

The strategy for categorizing decarbonization measures was temporal (before or after the Fourth IMO GHG Study 2020) rather than methodological (technical, operational, alternative fuel, or economical). $EI_{IMO}$ presented the increases in efficiency derived from the measures practiced before the Fourth IMO GHG Study 2020, as summarized in Section 2.2.

In detail, $Z_{OP}$ was estimated using the OP penetration rate ($\eta_{OP,t}$) and the decarbonization rate ($\alpha_{OP,t}$), as expressed in Equation (2). $\alpha_{OP,t}$ was set as 100% considering that carbon

emissions from electricity are assigned to the national budgets of coastal countries rather than the international maritime sector and, thus, $Z_{OP,t}$ was equalized to $\eta_{OP,t}$.

$$Z_{OP,t} = \eta_{OP,t} \times \alpha_{OP,t} \tag{2}$$

$Z_{ProCII,t}$ was determined on the basis of the gap between the actual emissions of ships assigned rates D and E ($e_{D\&E,t}$) and the supposed emissions if promoted to rate C ($e_{C,t}$). This was compared to the total emissions of the maritime sector ($e_{total}$), as expressed in Equation (3).

$$Z_{ProCII,t} = \sum (e_{D\&E,t} - e_{C,t})/e_{total} \tag{3}$$

The CII rating methods and boundaries were quoted from the IMO operational CII guidelines [30–33], which regulate the CII calculation methods, reference lines, reduction factors, and rating processes. As the CII guidelines entered into force in 2023, the CII rating promotion of the existing ships was considered. In this study, this was accomplished in the period of 2024–2030. To represent the 'existing ship' emissions, $e_{total}$ referred to the total emissions in 2020.

$Z_F$ was estimated using the zero-emission fuel penetration rate ($\eta_{i,j,t}$) and the fuel decarbonization rate ($\alpha_{i,j,t}$), as expressed in Equation (4):

$$Z_{F,t} = \sum_i \sum_j (\eta_{i,j,t} \times \alpha_{i,j,t}) \tag{4}$$

where $i$ and $j$ represent the fuel type and grade, respectively. Five types of alternative fuel including LNG, biodiesel, methanol, ammonia, and hydrogen were considered in the model. The fuel grades were set in accordance with the individual fuel sectors.

As expressed in Equation (5), the decarbonization rate of an individual fuel ($\alpha_i$) could be determined in the form of the carbon intensity ($CI_i$, $gCO_2/MJ$), using $CI_{LFO}$ as a reference.

$$\alpha_i = (1 - CI_i/CI_{LFO}) \tag{5}$$

In the calculations in this study, the consumption of pilot and blend fuels was taken into account in the forms of LFO/HFO and, thus, $\alpha_i$ represents the decarbonization potential of each alternative fuel/energy, regardless of the style of the propulsion system (internal combustion engine or fuel cell).

$CI_i$ could be determined using the lower calorific value (LCV, kJ/kg) and the conversion factor between fuel consumption and $CO_2$ emissions ($C_F$, $t\text{-}CO_2/t\text{-Fuel}$), as expressed in Equation (6).

$$CI_i = C_{F,i}/LCV_i \times 10^6 \tag{6}$$

*2.5. Parameterization*

When estimating $Z_{OP}$, the upper limit of $\eta_{OP,t}$ (equalized to $Z_{OP}$) was set as 12%, in accordance with the maximum share of practical energy consumption by an auxiliary engine.

The $Z_{ProCII}$ calculations were regulated by the IMO CII guidelines until 2026, with a yearly reduction factor of 2% from the rating baseline. For the years of 2027 to 2030, the reduction factors were set in accordance with the existing settings for 2024–2026 in this study.

As for $Z_F$, considering that the IMO LCA guidelines could determine the settings of the LCA emission factors until 2030, the parameterization in Equation (6) was in the TtW and WtW stages, respectively, before/during and after 2030. Before and during 2030, $C_{F,i}$ and $LCV_i$ were set in accordance with the TtW values set out in the IMO document [35], as listed in Table 4, where the values of $CI_i$ and $\alpha_i$ (calculated via Equation (6)) are also listed.

After 2030, $\alpha$ was estimated hierarchically in the WtW stage. The IMO excludes non-sustainable fuels from the candidate alternative fuel group, even if they are noncarbon fuels. Consequently, ships are supposed to use sustainable/renewable alternative fuels.

Table 5 summarizes the information related to possible sustainable fuels from the fuel and shipping sectors.

**Table 4.** TtW parameterizations of fuels.

| Type of Fuel | $LCV$ (kJ/kg) | $C_F$ (t-CO$_2$/t-Fuel) | $CI$ (gCO$_2$-eq/MJ) | $\alpha$ (%) |
|---|---|---|---|---|
| LFO | 41,200 | 3.151 | 76.5 | baseline |
| HFO | 40,200 | 3.114 | 77.5 | - |
| LNG | 48,000 | 2.75 | 57.3 | 25.1% |
| Methanol | 19,900 | 1.375 | 69.1 | 9.66% |
| Ammonia | 18,600 | 0 | 0 | 100% |
| Hydrogen | 120,000 | 0 | 0 | 100% |

**Table 5.** WtW parameterizations of fuels.

| | $CI$ (gCO$_2$-eq/MJ) [a] | $\alpha$ (%) [b] | References |
|---|---|---|---|
| LFO | 92 | baseline | thinkstep AG [36] |
| HFO | 90 | 2.17% | thinkstep AG [36] |
| e-LNG | 18 | 80.43% | thinkstep AG [36] |
| bio-LNG | 36 | 60.87% | IRENA [37] |
| UCO-biodiesel | 15 | 83.70% | ICCT [38] and adopted by IRENA [39] |
| e-methanol | 3 [c] | 96.74% | IRENA [37] |
| bio-methanol | 9 | 90.22% | Carlo Hamelinck [37] and adopted by IRENA [37,40] |
| e-ammonia | 19 [c] | 79.35% | IRENA [41] |
| bio-ammonia | 38 | 58.70% | IRENA [41] |
| hydrogen | 8 [c] | 90.98% | IEA [42] |

[a] $CI$ values were summarized from the references in the column on the right. [b] $\alpha$ values were calculated via Equation (5). [c] Data were comparable to those in the literature [43].

For LNG, methanol, and ammonia, e- and bio-fuels are deemed to be sustainable. e-fuels refer to fuels obtained using renewable CO$_2$ and H$_2$, as well as renewable energy. Bio-fuels refer to fuels produced from potential sustainable biomass feedstocks, including organic waste and by-products of industrial, agricultural, and municipal activities [36].

With regard to biodiesel, the IMO adopted a regulation to limit CI below 33 gCO$_2$-eq/MJ. Moreover, the feedstock should be sustainable. As such, biodiesel originating from recycled materials such as used cooking oil (UCO) best meets the IMO's specifications. According to an investigation by the International Council on Clean Transportation (ICCT), the CI of UCO biodiesel can be as low as 15 gCO$_2$-eq/MJ [38].

Low-emission hydrogen refers to products from water and renewable electricity (known as electrolysis), from fossil fuels with minimal CO$_2$ emissions (using carbon capture, utilization, and storage (CCUS)), and from bioenergy (e.g., via biomass gasification). The $CI$ of low-emission hydrogen was estimated to be ~8 gCO$_2$-eq/MJ in a production pathway using natural gas with CCUS with a 90% capture rate and renewable electricity [44].

Moreover, there were some principles, as listed below, for adjusting $Z_{OP}$, $Z_{ProCII}$, $\eta_{i,j,t}$, and $\alpha_{i,j,t}$ to make $Z_t$ meet the IMO's decarbonization goals.

Principle a: the scenario with lower transport growth (90% scenario) had slower on-shore power penetration, CII rating promotion, and e-fuel penetration and faster biodiesel penetration, which resulted in moderate efficiency improvements.

Principle b: the scenario with higher transport growth (130% scenario) had faster onshore power penetration, CII rating promotion, and alternative fuel penetration, which resulted in rapid efficiency improvements.

Principle c: to achieve the decarbonization goals in the 2023 IMO GHG Strategy, the penetration of alternative fuels could be set beyond the mean decarbonization levels of the fuels. This setting was reasonable considering that ship owners and captains have motivation to select sustainable alternative fuels registered in the IMO regulations. A similar situation was observed during the desulfurization of the shipping sector.

Principle d: before/during 2030, practical CII rating promotion was mainly achieved using biodiesel, and $Z_{ProCII}$ could be set based on the analysis of the IMO DCS data, while the penetration of biodiesel could be set as 0. After 2030, all measures could promote the CII rating simultaneously and $Z_{ProCII}$ could be set as 0 to avoid double counting.

## 3. Results and Discussions

### 3.1. Specification of IMO Decarbonization Goals

In this study, the CI requirements were used as a reference for the decarbonization pathway. Temporal CI reduction requirements were transformed from the reduction ambitions for the emission amount (Table 1) by combining them with the scenario-based projected $e_{CO2}$ (Tables 2 and 3). As shown in Figure 1, to achieve the IMO decarbonization goals for the emission amount (20–30%) in 2030, the reduction rates of CI needed to be as high as 51.5–57.7% and 57.0–62.5% in the 90% and 130% scenarios, respectively, which was much higher than the IMO-recommended value of 40%. With respect to the situation in 2040, the required CI reduction rates needed to be between 83.5% and 89.0% and between 87.0% and 91.5% in the 90% and 130% scenarios, respectively. Moreover, as the reduction goals for the emission amount were settled, the higher development rate in the 130% scenario resulted in a greater CI reduction requirement compared to the 90% scenario.

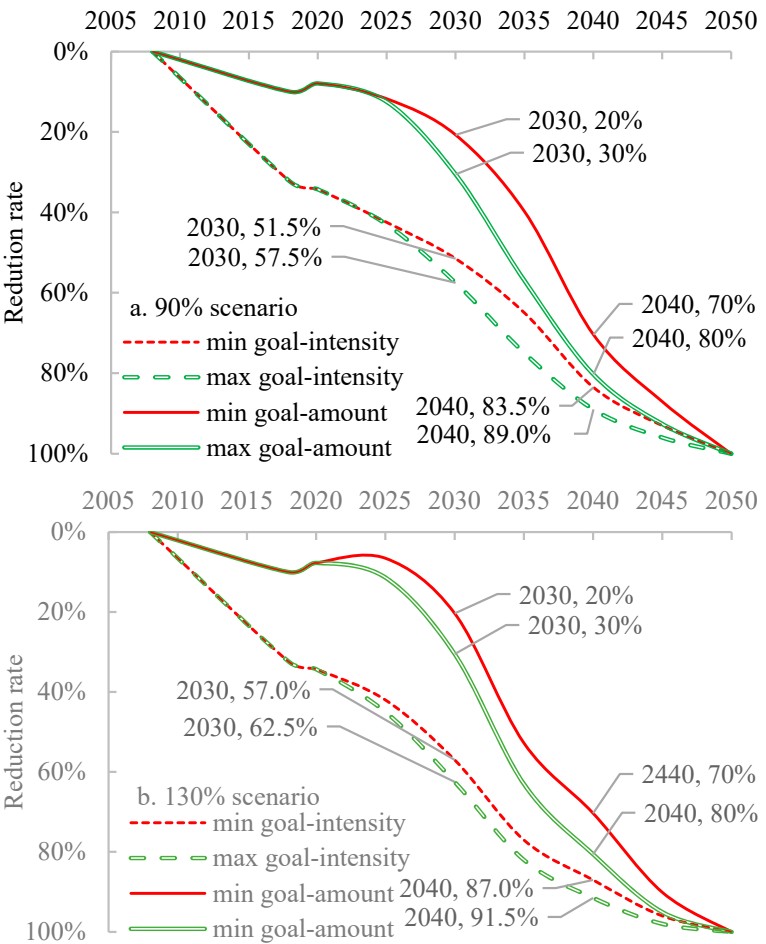

**Figure 1.** Temporal reduction requirements for emission intensity and amount.

### 3.2. Results of IMO DCS Data Analysis

Based on the 2020 IMO DCS data, the total reported $CO_2$ emissions of international shipping were determined to be $8.47 \times 10^9$ tonnes in 2020, representing a total fuel consumption of ~$2.72 \times 10^8$ tonnes HFO.eq. This statistic was low compared to the result predicted by the IMO ($1.02$–$1.03 \times 10^9$ tonnes, Tables 2 and 3), which could be attributed

to ruling out unregular data, as mentioned in Section 2.3. The portions of individual fuel/energy consumption (FC%) were also determined, and they are listed in Table 6, showing that HFO and LFO were still the dominant marine fuels. The usage of onshore power was very limited globally, which was consistent with the observations in the literature [17,45] and implied considerable decarbonization potential. These portions were used as references when setting the fuel/energy penetration in the decarbonization pathway.

**Table 6.** Portion of fuel/energy consumption.

|  | FC% |
|---|---|
| LFO | 39.1% |
| HFO | 60.6% |
| LNG | 0.30% |
| Biodiesel | 0.03% |
| Methanol | 0.02% |
| Ammonia | <0.01% |
| Hydrogen | <0.01% |
| Onshore power | <0.01% |

According to the 2020 IMO DCS data, 6.01% and 6.09% of the $CO_2$ emissions would be reduced if all ships with D and E classes were promoted to the C class in the years 2025 and 2030, respectively. These results were used as the upper limit of $Z_{ProCII}$.

### 3.3. Technical Requirements

The technical requirements were determined by adjusting the decarbonization potentials of individual measures to make Z fall in the scope of the decarbonization goals in the form of the CI specified in Section 3.1. The setups of the decarbonization pathway are illustrated in Figure 2. The results show that the application of sustainable alternative fuels was the most crucial way to achieve the decarbonization goals. Nevertheless, zero emissions are hardly likely to be achieved until 2050 due to the unavailability of net-zero-emission fuels (Table 5).

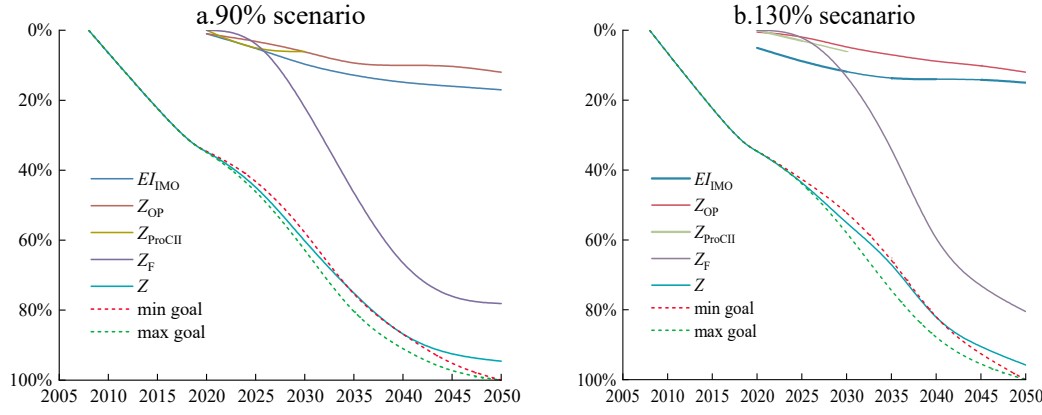

**Figure 2.** Requirements of carbon intensity of various methods in decarbonization pathway.

In 2030, $Z_{OP}$ and $Z_{ProCII}$ were set as 5.0% and 6.09% (Tables 7 and 8), respectively, in both scenarios. $Z_{OP}$ was set at less than the medium value of the maximum reduction potential (12%), considering that the growth of onshore power might be at a moderate speed due to the absence of a related IMO regulation before 2030. After 2030, $Z_{OP}$ could grow continuously with the scenario's development. By contrast, $Z_{ProCII}$ was set as the upper limit value in 2030 due to the execution of IMO operational CII rating regulations since 2023.

**Table 7.** Parameterizations in decarbonization pathway in 90% scenario.

|  | $EI_{\text{IMO}}$ | $Z_{\text{OP}}$ | $Z_{\text{ProCII}}$ | $Z_{\text{F}}$ | $Z$ |
|---|---|---|---|---|---|
| 2008 | null | null | null | null | null |
| 2018 | null | null | null | null | 32.38% |
| 2020 | 5.0% | 0.5% | 0% | 0.00% | 34.26% |
| 2025 | 9.0% | 1.5% | 3.01% | 0.26% | 43.03% |
| 2030 | 12.0% | 5.0% | 6.09% | 11.78% | 55.38% |
| 2035 | 14.0% | 7.0% | null [a] | 32.51% | 65.80% |
| 2040 | 14.0% | 9.0% | null [a] | 62.69% | 84.00% |
| 2045 | 14.0% | 10.0% | null [a] | 73.79% | 90.84% |
| 2050 | 15.0% | 12.0% | null [a] | 80.51% | 95.82% |

[a] Refer to parameterization 'principle d' in Section 2.5.

**Table 8.** Parameterizations in decarbonization pathway in 130% scenario.

|  | $EI_{\text{IMO}}$ | $Z_{\text{OP}}$ | $Z_{\text{ProCII}}$ | $Z_{\text{F}}$ | $Z$ |
|---|---|---|---|---|---|
| 2008 | null | null | null | null | null |
| 2018 | null | null | null | null | 32.38% |
| 2020 | 1.00% | 1.0% | 0.00% | 0.00% | 34.31% |
| 2025 | 5.00% | 3.0% | 6.01% | 0.55% | 43.56% |
| 2030 | 10.0% | 6.0% | 6.09% | 21.16% | 60.54% |
| 2035 | 13.0% | 10% | null [a] | 47.19% | 75.54% |
| 2040 | 15.0% | 10% | null [a] | 68.65% | 88.08% |
| 2045 | 16.0% | 10% | null [a] | 77.37% | 93.03% |
| 2050 | 17.0% | 12% | null [a] | 78.08% | 94.59% |

[a] Refer to parameterization 'principle d' in Section 2.5.

With regard to $Z_{\text{F}}$, it had to be 11.8% and 21.16%, respectively, to meet the decarbonization goals in the 90% and 130% scenarios by 2030. By 2040, $Z_{\text{F}}$ had to be 62.7% and 68.7%, respectively, in the 90% and 130% scenarios. $Z_{\text{F}}$ was required to grow rapidly in the 2030s, accounting for 50.9% and 47.5% in the 90% and 130% scenarios, respectively, which indicated an urgent need for the shipping sector to switch to renewable zero-emission fuels in this time period. These results are generally in accordance with a recent prediction in the literature [46]. The gap in the required $Z_{\text{F}}$ between the 90% and 130% scenarios is much greater in 2030 and 2035 compared to 2040 and later, which indicates that higher transport growth will likely result in a larger share of renewable fuels in the 2030s. By 2050, $Z_{\text{F}}$ is required to achieve ~95% in both scenarios.

To achieve the required $Z_{\text{F}}$, the alternative fuel penetrations were set as shown in Figure 3. The total penetration ($\eta_{\text{total}}$) values needed to be 15.0% and 26.0% in 2030 to achieve the decarbonization goals in the 90% and 130% scenarios, respectively. These requirements were both far beyond the expected value in the 2023 IMO GHG Strategy (i.e., 5–10%, Table 1). By 2040, the $\eta_{\text{total}}$ values were required to be 76.0% and 82.0% in the 90% and 130% scenarios, respectively.

It is projected that ~40 million GTs of new ships will be delivered annually [47], accounting for ~2.5% of the global fleet (~1600 million GTs) [48]. At this level, provided that all new ships are powered by alternative fuels, alternative-fuel ships will account for ~20% and ~45% of the global fleet by 2030 and 2040, respectively. These shares are close to the required $\eta_{\text{total}}$ values, especially in 2040, indicating that the shipping sector is likely under pressure to promote the use of alternative-fuel ships.

Furthermore, high proportions of e-fuels were required (Figures 4 and 5), especially after 2035. In the 130% scenario, the e-fuel requirements were as high as 50% and 75%, respectively, in 2030 and 2040. By 2050, the proportion had to be 95% for e-methanol (eMeOH) and e-ammonia (eNH$_3$). Even in the 90% scenario, the proportion of e-methanol and e-ammonia needed to reach 60% and 85%, respectively, by 2040 and 2050. These proportions were much higher than the predicted development of e-fuel supplies [49], indicating a higher fuel price for the shipping sector compared to the average social level.

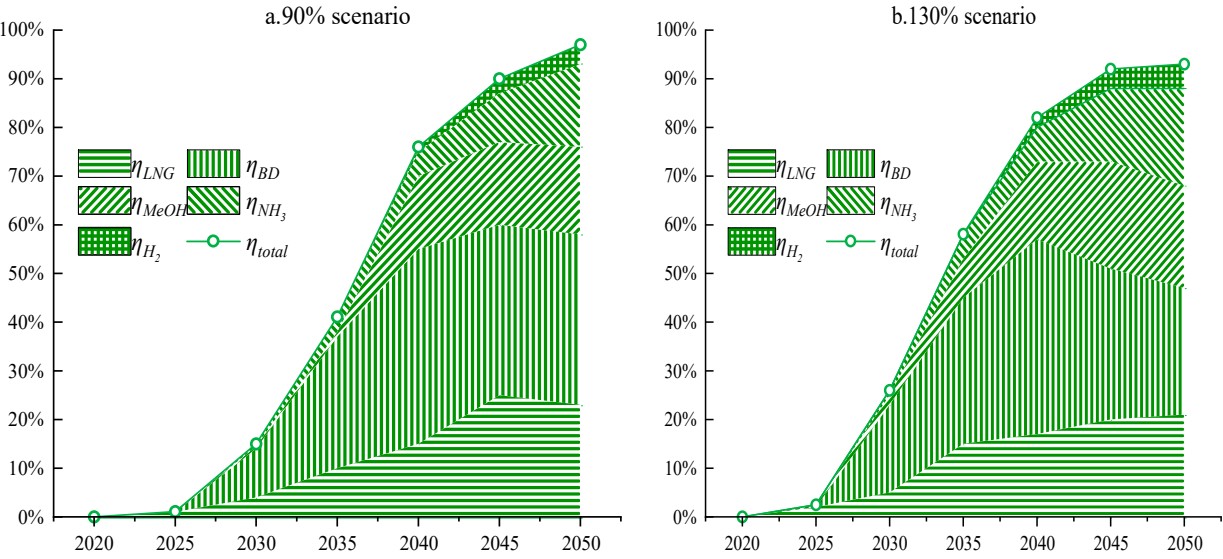

**Figure 3.** Requirements for alternative fuel penetration.

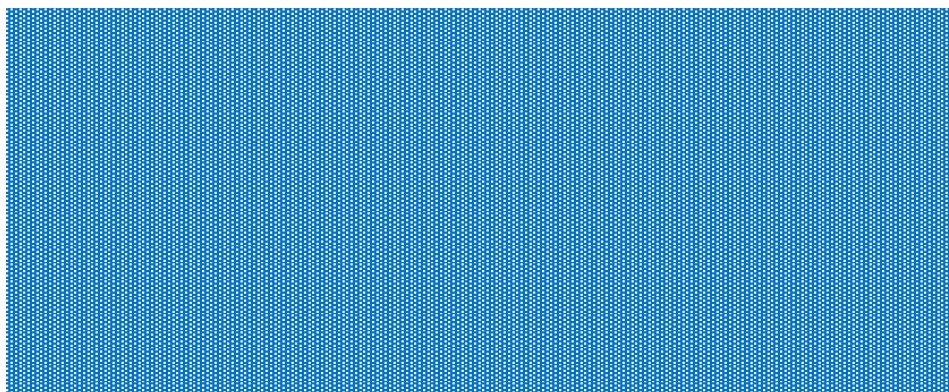

**Figure 4.** Requirements for fuel grade composition in 90% scenario.

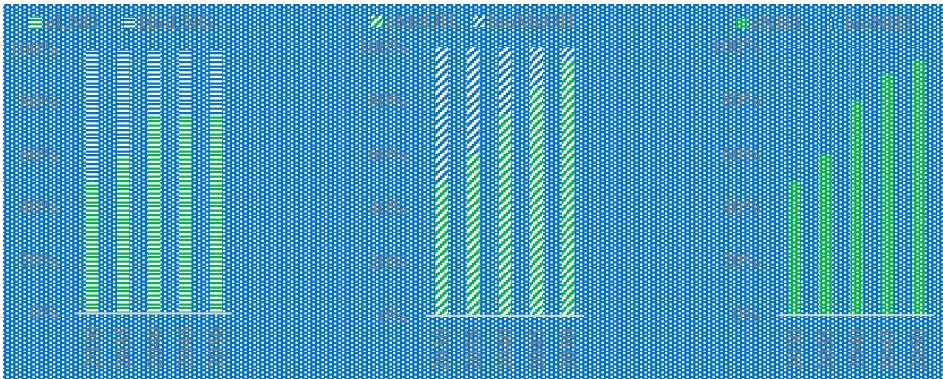

**Figure 5.** Requirements for fuel grade composition in 130% scenario.

Compared to research before the Fourth IMO GHG Study 2020, the requirements to achieve the IMO decarbonization goals were more stressful in this study. In this period, the parameterizations of the technical and operational measures were set as over 40% in 2035 or later [16,50], regardless of the limits of speed reduction, payload utilization, and technical improvements to existing ships. Moreover, the inherent growth of international shipping has been ignored in the literature, which has likely resulted in a huge underestimation of the requirements. In fact, the decarbonization requirements were much higher than the recommended '40% in intensity', as specified in Figure 1.

### 3.4. Weight Analysis of Methods

The weights of the various measures were analyzed, referring to the results of a sensitivity analysis via Monte Carlo simulation using Oracle Crystal Ball (version 11.1.3.0.0) software, as shown in Figure 6. In the short and medium terms (2030), when the share of the sustainable fuels was minor, pathways including fleet composition, regulatory efficiency improvements, market-driven efficiency improvements, and onshore power usage were important, judging from the high variance contributions of $EI_{IMO}$ and $Z_{OP}$.

Combining Figures 2–6, it can be observed that the sensitivities of the parameters were positively correlated with their valuations. $\eta_{BD}$, $\eta_{LNG}$, and $\alpha_{BD}$ were quite sensitive due to their high penetrations, especially before 2040, which reflected that the usages of biodiesel and LNG had important roles in achieving the decarbonization goals. By contrast, methanol and ammonia were less important due to their lower penetrations. These settings were reasonable considering that the global fleet cannot eliminate all its ships powered by conventional fuels, even if all new ships use alternative fuels (i.e., methanol and ammonia). In reality, new building orders in 2023 are still dominated by ships that use conventional fuels, accounting for 55% (GT%) of all orders. In orders for alternative-fuel ships, the largest share is attributed to LNG dual-fuel ships, which account for 25% (GT%) [51]. In addition, the contribution of hydrogen is minor due to its immaturity in safe usage, storage, and conversion onboard [52], while the importance of hydrogen may be reflected by its usage as the precursor of e-fuels.

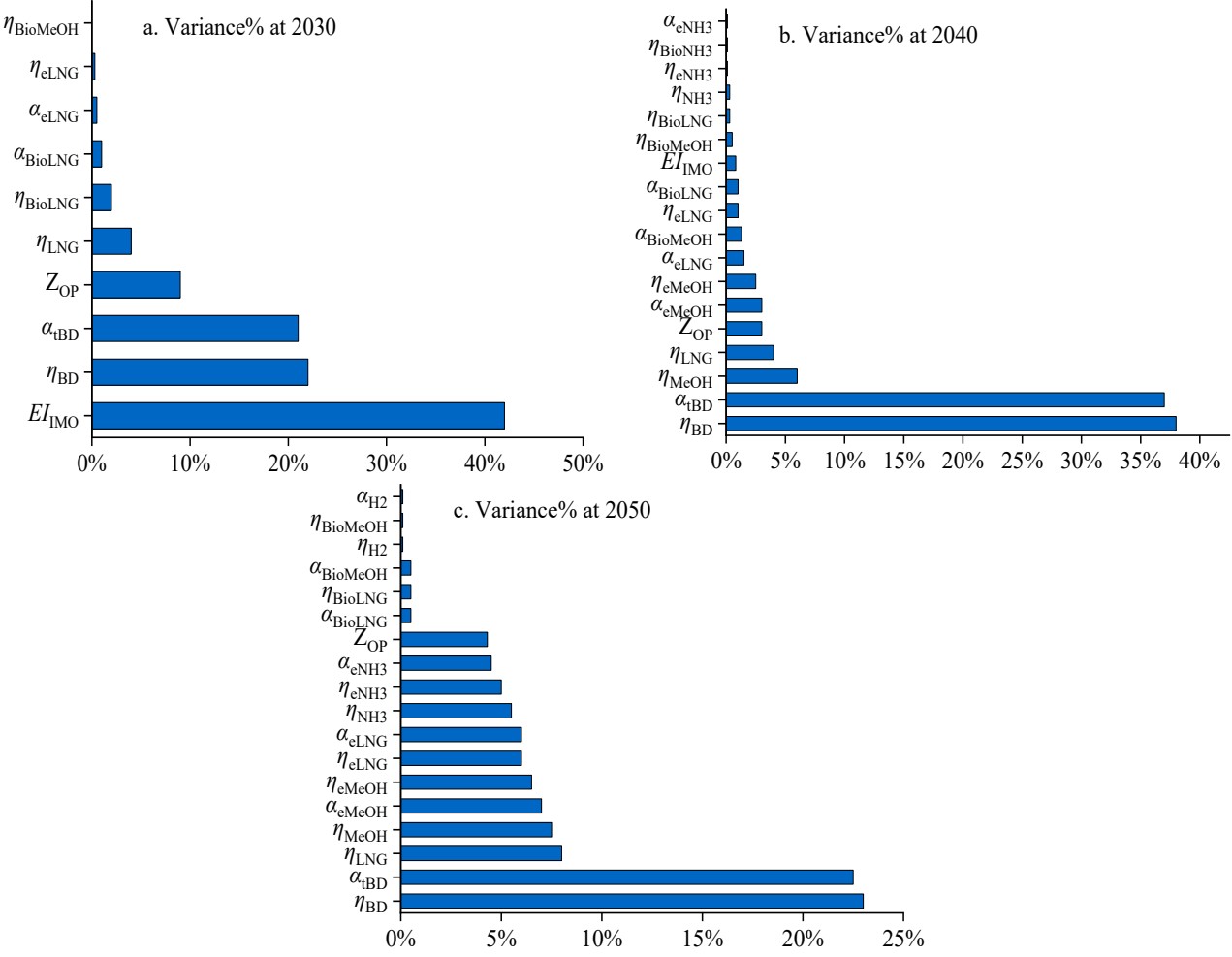

**Figure 6.** Results of sensitivity analysis of the decarbonization model.

### 4. Conclusions

This study is the first quantitative interpretation of the target in the 2023 IMO GHG Strategy. The integral decarbonization of short-term measures suggested by the IMO, annual IMO DCS data, and the LCA decarbonization potential of alternative fuels were applied for the first time in the predictions.

The decarbonization goals for emission intensity are actually 51.5–62.5% in the IMO GHG reduction scenarios. This means that a deviated prediction will be made if the 'apparent' IMO-recommended value of 40% is set as the target. On the basis of the rational assessment of the decarbonization potentials of measures by the IMO and the energy sector, the intense technical requirements behind the IMO's ambitions were revealed. Combined with the continuous application of short-term measures, onshore power and regulations will be required to contribute their maximum potential no later than 2030. Even so, considerable penetration (15.0–26.0%) of alternative fuels will be required in 2030 to achieve the decarbonization goals in the 90% and 130% scenarios, respectively, and both values are far beyond the expected value in the 2023 IMO GHG Strategy (i.e., 5–10%). High proportions of e-fuels will be required, especially after 2035, reaching as high as 85–95% for e-methanol and e-ammonia by 2050. Sustainable grades of biodiesel and LNG will also be necessary choices for the decarbonization of the shipping sector. The role of hydrogen in the shipping sector might be as the precursor of e-fuels.

Overestimations of individual measures and their integral decarbonization potentials, as well as favorable fuel suggestions, were avoided in this study in order to make sensible predictions in accordance with the strategy principles. Thus, the technical requirements represent a fair and just transition pathway to achieve the target of the 2023 IMO GHG Strategy. Although the proposed decarbonization pathway might not be the only option for the shipping sector, the findings of this study highlight the stressful technical requirements behind the 2023 IMO GHG Strategy and the importance of taking appropriate actions.

As all the technical requirements were at or beyond the upper limits of the technical decarbonization potential and the average social level, decarbonization in the shipping sector will occur at a higher cost compared to the social average. Therefore, economic factors will be necessary to affect the implementation of the strategy. A framework of economic measures was proposed in the 2023 IMO GHG Strategy to offset the decarbonization cost, and further techno-economic assessments will be necessary in future studies. In addition, the emissions of methane and nitrous oxide were not included in this study due to a lack of relevant information. The model will be improved once these data are ready, in accordance with the IMO LCA guidelines.

**Author Contributions:** C.Z.: Organization and Writing. J.Z.: Data curation and Writing—original draft preparation. H.G.: Data collection and Validation. S.X.: Policy analysis. X.W.: Policy analysis. Z.W.: Data collection. T.C.: Data collection. L.Y.: Data cleaning. X.Z.: Algorithm support. P.S.: Organization and Methodology. All authors have read and agreed to the published version of the manuscript.

**Funding:** This research was funded by the National Key R&D Program of China (2022YFB4301403).

**Institutional Review Board Statement:** Not applicable.

**Informed Consent Statement:** Not applicable.

**Data Availability Statement:** Fuel oil consumption data is available one the IMO's website: https://gisis.imo.org/Members/FUEL/ShipsList.aspx. Register for free access to resources made available to the public by IMO.

**Conflicts of Interest:** The authors declare no conflicts of interest.

## Abbreviations

| | |
|---|---|
| BAU | Business-as-usual |
| BD | Biodiesel |
| CCUS | Carbon capture utilization and storage |
| $C_F$ | Fuel consumption |
| CI | Carbon intensity |
| CII | Carbon intensity indicator |
| DCS | Data collection system |
| $e_{CO2}$ | Emission of $CO_2$ |
| EEDI | Energy efficiency design index |
| EFs | Emission factors |
| $EI_{IMO}$ | Efficiency improvements |
| eMeOH | e-methanol |
| $eNH_3$ | e-ammonia |
| GHG | Greenhouse gas |
| GT | Gross tonne |
| HFO | Heavy fuel oil |
| ICCT | International Council on Clean Transportation |
| IMO | International Maritime Organization |
| IPCC | Intergovernmental Panel on Climate Change |
| LCA | Life cycle assessment |
| LFO | Light fuel oil |
| LNG | Liquified natural gas |
| MARPOL | Maritime Agreement Regarding Oil Pollution |
| MEPC | Maritime Environment Protection Committee |
| OP | Onshore power |
| ProCII | CII rating promotion |
| RCPs | Representative concentration pathways |
| TtW | Tank-to-wake |
| UOC | Used cooking oil |
| WtW | Well-to-wake |

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
