# Peer review of "Technical Requirements for 2023 IMO GHG Strategy"

_sustainability, doi:10.3390/su16072766_

Round 1
Reviewer 1 Report
Comments and Suggestions for Authors
Review-MDPI
Place |
It is |
I propose |
Remark |
Page 2 Line 51 |
…that and more than half of… |
…that more than half of… |
|
Page 2 Line |
…is expected to further push forward… |
…is expected to produce a further push forward |
|
Page 2 Line 73 |
…to determined or modeled. |
…to determine or model. |
|
Page 2 Line 80 |
…are not taken account into the national carbon budget… |
…are not considered by the national carbon budget… |
|
Page 2 Line 85 |
…model [40] which likely exhibits a great uncertainty. |
…model [40] which will likely exhibit a great uncertainty. |
|
Page Line 99 |
Parametrizations… |
Parameterizations… |
|
Page 3 104 |
In this study, IMO ambitions in and indicative checkpoints 2023 IMO GHG strategy were summarized… |
In this study, IMO ambitions and indicative checkpoints 2023 IMO GHG strategy were summarized… |
|
Page 3 Line 114 |
… will be ad power usage ted for shipping that has an impaction emissions or energy efficiency. |
|
I do not understand the sentence… |
Page 3 Line 116 |
In the context of this study, analysis would be proceeded on the basis of the data in 90% and 130% scenarios as summarized in… |
In the context of this study, analysis will proceed on the basis of the data in 90% and 130% scenarios as summarized in… |
|
Page 5 Line 181 |
…action on onshore power usage, CII promoting and fuel alteration has… |
…action on onshore power usage; CII promoting and fuel alteration has… |
|
Page 5 Line 16-198 |
|
|
I do not understand the sentence… |
Page 5 Line 202-204 |
As CII guidelines has entered into force at 2023, the CII rating promotion of the existing ship was assessed to accomplished in the period of 2024-2030 in this study. |
As CII guidelines has entered into force at 2023, the CII rating promotion of the existing ship was considered, in this study, that will be accomplished in the period of 2024-2030. |
Is my interpretation correct? I am not sure. Clarify, please. |
Page 5 Line 209 |
Where i and j represented… |
Where i and j represent… |
|
Page 5 Line 216 |
…represented the decarbonization… |
…represent the decarbonization… |
|
Page 5 Line 220 |
…as expressed by… |
…is expressed by… |
|
Page 5 Line 225 |
…with the max… |
…with the maximum… |
|
Page 6 Line 230 |
…parametrization… |
…parameterization… |
|
Page 6 Line 251 |
With regard to the biodiesel, IMO has adopted a regulation [54] to restrict that CI shall be less than 33 gCO2-eq/MJ. |
With regard to biodiesel, IMO has adopted a regulation [54] to limit CI below 33 gCO2-eq/MJ. |
|
Page 7 Line 277 |
|
|
I do not understand the sentence… |
Page 8 Line 304 |
…was very limit… |
…was very limited… |
|
Page 8 Line 316 |
…was hardly to be achieved… |
…is hardly to be achieved… |
|
Page 9 Line 328 |
…was required… |
…is required… |
|
Page 9 Line 331 |
…were generally… |
…are generally… |
|
Page 9 Line 332 |
…was much greater… |
…is much greater… |
|
Page 9 Line 333 |
…which indicated… |
…which indicate… |
|
Page 9 Line 334 |
…likely resulted… |
…will likely result… |
|
Page 9 Line 335 |
…was required… |
…is required… |
|
Page 12 Line 389 |
…considering that the global fleet is impossible to get rid of all convention fuel powered ships even if all the new buildings are alternative fuel (i.e. methanol and ammonia) powered. |
…considering that the global fleet cannot get rid of all its conventional fuel powered ships, even if all the new ships will use alternative fuel (i.e. methanol and ammonia). |
Is my interpretation correct? I am not sure. Clarify, please. |
Page 13 Line 403 |
…parametrization… |
…parameterization… |
|
Page 13 Line 405-411 |
|
|
Improve the English. |
Page 13 Line 412 |
…was very limit… |
…was very limited… |
|
Page 13 Line 413 |
…there was… |
…there will be … |
|
Page 13 Line 413 |
…alternative fuels was… |
…will be… |
|
Page 13 Line 423 |
…were required… |
…will be required… |
|
Overall, the article is interesting and actual, mostly if the actual model of globalization of trade and sea transport remains without improvements.
The most serious problem is that, in general, it is not clear the way/processes/algorithms used to perform the obtain the results referred in “Results and discussions”. The paper must be improved by summary and rigorous indication of these calculations.
A scrupulous review of the text is necessary. I believe that I have signaled some of these flaws in the previous table.
See format of Tables 2 and 3: correct the misalignments.
More than 40% of references were published less than 5 years ago, which ensures the article is up to date.
Add to Abbreviations: MARPOL, eCO2, CF, CI, UCO, etc. Add a list with Greek letter: a, etc.
Comments on the Quality of English Language
Review-MDPI
Place |
It is |
I propose |
Remark |
Page 2 Line 51 |
…that and more than half of… |
…that more than half of… |
|
Page 2 Line |
…is expected to further push forward… |
…is expected to produce a further push forward |
|
Page 2 Line 73 |
…to determined or modeled. |
…to determine or model. |
|
Page 2 Line 80 |
…are not taken account into the national carbon budget… |
…are not considered by the national carbon budget… |
|
Page 2 Line 85 |
…model [40] which likely exhibits a great uncertainty. |
…model [40] which will likely exhibit a great uncertainty. |
|
Page Line 99 |
Parametrizations… |
Parameterizations… |
|
Page 3 104 |
In this study, IMO ambitions in and indicative checkpoints 2023 IMO GHG strategy were summarized… |
In this study, IMO ambitions and indicative checkpoints 2023 IMO GHG strategy were summarized… |
|
Page 3 Line 114 |
… will be ad power usage ted for shipping that has an impaction emissions or energy efficiency. |
|
I do not understand the sentence… |
Page 3 Line 116 |
In the context of this study, analysis would be proceeded on the basis of the data in 90% and 130% scenarios as summarized in… |
In the context of this study, analysis will proceed on the basis of the data in 90% and 130% scenarios as summarized in… |
|
Page 5 Line 181 |
…action on onshore power usage, CII promoting and fuel alteration has… |
…action on onshore power usage; CII promoting and fuel alteration has… |
|
Page 5 Line 16-198 |
|
|
I do not understand the sentence… |
Page 5 Line 202-204 |
As CII guidelines has entered into force at 2023, the CII rating promotion of the existing ship was assessed to accomplished in the period of 2024-2030 in this study. |
As CII guidelines has entered into force at 2023, the CII rating promotion of the existing ship was considered, in this study, that will be accomplished in the period of 2024-2030. |
Is my interpretation correct? I am not sure. Clarify, please. |
Page 5 Line 209 |
Where i and j represented… |
Where i and j represent… |
|
Page 5 Line 216 |
…represented the decarbonization… |
…represent the decarbonization… |
|
Page 5 Line 220 |
…as expressed by… |
…is expressed by… |
|
Page 5 Line 225 |
…with the max… |
…with the maximum… |
|
Page 6 Line 230 |
…parametrization… |
…parameterization… |
|
Page 6 Line 251 |
With regard to the biodiesel, IMO has adopted a regulation [54] to restrict that CI shall be less than 33 gCO2-eq/MJ. |
With regard to biodiesel, IMO has adopted a regulation [54] to limit CI below 33 gCO2-eq/MJ. |
|
Page 7 Line 277 |
|
|
I do not understand the sentence… |
Page 8 Line 304 |
…was very limit… |
…was very limited… |
|
Page 8 Line 316 |
…was hardly to be achieved… |
…is hardly to be achieved… |
|
Page 9 Line 328 |
…was required… |
…is required… |
|
Page 9 Line 331 |
…were generally… |
…are generally… |
|
Page 9 Line 332 |
…was much greater… |
…is much greater… |
|
Page 9 Line 333 |
…which indicated… |
…which indicate… |
|
Page 9 Line 334 |
…likely resulted… |
…will likely result… |
|
Page 9 Line 335 |
…was required… |
…is required… |
|
Page 12 Line 389 |
…considering that the global fleet is impossible to get rid of all convention fuel powered ships even if all the new buildings are alternative fuel (i.e. methanol and ammonia) powered. |
…considering that the global fleet cannot get rid of all its conventional fuel powered ships, even if all the new ships will use alternative fuel (i.e. methanol and ammonia). |
Is my interpretation correct? I am not sure. Clarify, please. |
Page 13 Line 403 |
…parametrization… |
…parameterization… |
|
Page 13 Line 405-411 |
|
|
Improve the English. |
Page 13 Line 412 |
…was very limit… |
…was very limited… |
|
Page 13 Line 413 |
…there was… |
…there will be … |
|
Page 13 Line 413 |
…alternative fuels was… |
…will be… |
|
Page 13 Line 423 |
…were required… |
…will be required… |
|
Overall, the article is interesting and actual, mostly if the actual model of globalization of trade and sea transport remains without improvements.
The most serious problem is that, in general, it is not clear the way/processes/algorithms used to perform the obtain the results referred in “Results and discussions”. The paper must be improved by summary and rigorous indication of these calculations.
A scrupulous review of the text is necessary. I believe that I have signaled some of these flaws in the previous table.
See format of Tables 2 and 3: correct the misalignments.
More than 40% of references were published less than 5 years ago, which ensures the article is up to date.
Add to Abbreviations: MARPOL, eCO2, CF, CI, UCO, etc. Add a list with Greek letter: a, etc.
Reviewer 2 Report
Comments and Suggestions for Authors
1- The abstract should be rewritten to show the objectives, main findings and comparative values. The abstract should be enhanced by providing a more comprehensive explanation of the uniqueness of your article.
2- The introduction is weak and should be supported with other references.
3- The novelty of paper should be focused.
4- The assumptions of model should be stated.
5- The discussions of the results should be supported.
6- Discuss the limitations of the study
7- The manuscript's conclusion could be more explicit. What are the main findings of the study? What are the implications of these findings for future research in the field?
8- The references should be updated .
The paper should be accepted with major revision.
Comments on the Quality of English LanguageThe structure and grammar should be revised.
Reviewer 3 Report
Comments and Suggestions for Authors
1. The research content of this paper is closely related to IMO ambitions in and indicative checkpoints 2023 IMO GHG strategy, and the results can be a reference for IMO to implement GHG strategy.
2. The model of the total decarbonization potential in emission intensity (Zt, %) (Eq.1) Seems a little simple because a few factors. A mathematical model based on these limited parameters can only be used as a general reference.
3. This paper studies the Technical requirements for 2023 IMO GHG Strategy, but what are the Technical requirements need to be clearly pointed out.
4. The Introduction of this paper does not introduce relevant international studies, nor does it compare with other relevant studies in this study, so it is impossible to know the superiority of this study. 64 references are listed in this paper, but the citation analysis of these references is not deep enough, and there is a lack of references compared with the research.
Round 2
Reviewer 1 Report
Comments and Suggestions for Authors
No more comments.
Reviewer 2 Report
Comments and Suggestions for Authors
I think the comments were considered by the authors. The manuscript should be accepted.